# Circulating Small Extracellular Vesicle-Derived miR-342-5p Ameliorates Beta-Amyloid Formation via Targeting Beta-site APP Cleaving Enzyme 1 in Alzheimer’s Disease

**DOI:** 10.3390/cells11233830

**Published:** 2022-11-29

**Authors:** Zhiwu Dong, Hongjun Gu, Qiang Guo, Xianglu Liu, Feifei Li, Huiling Liu, Li Sun, Huimin Ma, Kewen Zhao

**Affiliations:** 1Department of Laboratory Medicine, Jinshan Branch of Shanghai Sixth People’s Hospital, Jinshan District Central Hospital Affiliated to Shanghai University of Medicine & Health Sciences, Shanghai 201599, China; 2Department of Geriatric Medicine, Shanghai Jinshan District Hospital of Integrated Traditional Chinese and Western Medicine, Shanghai 201501, China; 3Department of Ultrasound Medicine, Jinshan Branch of Shanghai Sixth People’s Hospital, Jinshan District Central Hospital affiliated to Shanghai University of Medicine & Health Sciences, Shanghai 201599, China; 4Department of Pathophysiology, Shanghai Jiaotong University School of Medicine, Shanghai 200025, China

**Keywords:** extracellular vesicle, miRNA, Alzheimer’s disease, BACE1, amyloid-beta, neurons

## Abstract

Alzheimer’s disease (AD) is a common neurodegenerative disorder with progressive cognitive impairment in the elderly. Beta-amyloid (Aβ) formation and its accumulation in the brain constitute one of the pathological hallmarks of AD. Until now, how to modulate Aβ formation in hippocampal neurons remains a big challenge. Herein, we investigated whether the exosomal transfer of microRNA (miR) relates to amyloid pathology in the recipient neuron cells. We isolated circulating small extracellular vesicles (sEVs) from AD patients and healthy controls, determined the miR-342-5p level in the sEVs by RT-PCR, and evaluated its diagnostic performance in AD. Then, we took advantage of biomolecular assays to estimate the role of miR-342-5p in modulating the amyloid pathway, including amyloid precursor protein (APP), beta-site APP cleaving enzyme 1 (BACE1), and Aβ42. Furthermore, we subjected HT22 cells to the sEVs from the hippocampal tissues of transgenic APP mice (Exo-APP) or C57BL/6 littermates (Exo-CTL), and the Exo-APP enriched with miR-342-5p mimics or the control to assess the effect of the sEVs’ delivery of miR-342-5p on Aβ formation. We observed a lower level of miR-342-5p in the circulating sEVs from AD patients compared with healthy controls. MiR-342-5p participated in Aβ formation by modulating BACE1 expression, specifically binding its 3′-untranslated region (UTR) sequence. Exo-APP distinctly promoted Aβ42 formation in the recipient cells compared to Exo-CTL. Intriguingly, miR-342-5p enrichment in Exo-APP ameliorated amyloid pathology in the recipient cells. Our study indicated that miR-342-5p was dysregulated in human circulating sEVs from AD patients; sEV transfer of miR-342-5p ameliorates Aβ formation by modulating BACE1 expression. These findings highlight the promising potential of exosomal miRNAs in AD clinical therapy.

## 1. Introduction

Alzheimer’s disease (AD) is anatomically characterized by neocortical atrophy, neuron and synapse loss, and principal pathological features including extracellular plaques of beta-amyloid (Aβ) deposits and intracellular neurofibrillary tangles (NFTs). The mechanisms underlying the origin of AD and its progression remain imperfectly understood. In light of many failures in the clinical treatment and drug development, there is an urgent need to identify potential therapeutic targets for prodromal and established AD [1].

Aβ deposition is considered a main pathogenic event in Alzheimer’s disease, forming oligomers as major constituents of the senile plaques. Abnormal Aβ production and accumulation at nerve terminals, such as presynaptic terminals in the dentate gyrus, lead to synaptic pathology and ultimately neurodegeneration [2]. Aβ production from amyloid precursor protein (APP) depends on the sequential proteolysis by the beta-site APP-cleaving enzyme (BACE, also known as β-secretase) and γ-secretase [3]. BACE1, γ-secretase, and APP constitute a molecular complex in vivo, facilitating APP shuttling and processing [4]. Conversely, growing evidence suggests a physiological role for Aβ in the central nervous system, participating in neurogenesis [5], neuron survival [6], neuron electrophysiology [7], neuron transmission [8], synaptic plasticity [9], learning and memory [10], hippocampal memory consolidation [11], and long-term potentiation [12]. Furthermore, it was presumed that the transient increase in Aβ serves as an adaptive response to counteract a variety of challenges or insults upon neurons, including chronic inflammation [13], bacterial infection [14], and oxidative stress [15].

Increased BACE1 activity plays a key role in neurodegeneration in AD brains, although its molecular alterations remain ambiguous. Human BACE1 knock-in sufficiently elicited degeneration of neurons in the neocortex and hippocampus in a transgenic mouse model, independent of Aβ production and even APP level [16]. Partial reduction of BACE1 led to only a 12% decrease in Aβ level, obvious reduction of Aβ deposition, and synaptic deficits in a transgenic mouse AD model overexpressing human mutated APP [17]. Partial reduction of BACE1 decreased the levels of APP-βCTF in the brain, instead of Aβ or APP, preventing neuronal endosomal pathology, loss of neurons, and development of the AD-related phenotype in Ts2.BACE1^+/−^ mice [18]. Thus, how to efficaciously control the BACE1 activity that associates with pathological Aβ production is still a pending problem.

The dysregulation of microRNAs (miRNAs) plays an important role in the pathogenesis of AD [19]. Intriguingly, recent work revealed that miR-342-5p was profoundly elevated in the brains of one- and three-month-old APP and presenilin 1 (PS1) double transgenic mice compared with age-matched wild-type littermates [20]. Previous work demonstrated that miR-342-5p was upregulated in the hippocampi of APP/PS1, PS1DeltaE9, and PS1-M146V AD transgenic mice, contributing to AD axonopathy [21]. Furthermore, upregulation of miR-342-5p may downregulate the expression of ankyrin G (AnkG) and is involved in defective selective filtering at the axon initial segment (AIS) in cultured hippocampal neurons from the AD mouse model [22]. On the contrary, a series of miRNAs in plasma exosomes showed significant differences in AD patients, including miR-342-5p, which was downregulated in the AD exosomes [23]. In light of the opposite trends of miR-342-5p in circulating exosomes versus the hippocampus, we performed this study to investigate the roles of circulating sEV derived miR-342-5p in AD pathology.

## 2. Materials and Methods

### 2.1. Subjects and Sample Preparation

This study was approved by the Ethics Committee of Jinshan Branch of Shanghai Sixth People’s Hospital and complied with the World Medical Association Declaration of Helsinki regarding the ethical conduct of research involving human subjects. AD was diagnosed according to the recommendations from the National Institute on Aging–Alzheimer’s Association workgroups [24]. AD Patients were recruited to the study from Shanghai Jinshan Zhongren Aged Care Hospital and Jinshan Branch of Shanghai Sixth People’s Hospital from October 2016 to December 2019. The severity of dementia was assessed by a minimental state examination (MMSE) score. Healthy controls were age-matched, nondemented outpatients without cognitive problems for physical examination. Those with malignant tumors, a history of cerebrovascular diseases, ischemic or hemorrhagic stroke, severe metabolic diseases, and severe liver and kidney dysfunctions were excluded from this study. Informed consent was obtained from all the AD patients or their legal representatives as well as the control subjects. Details of the subject demographics are shown in Table 1. Fasting venous blood samples were collected from the participants in the morning. The serum was separated within two hours by centrifuging at 1500× *g* for 10 min at room temperature. Then the samples were centrifuged at 16,000 rpm for 5 min at 4 °C. The supernatant was stored at −80 °C for later use.

### 2.2. Isolation and Characterization of Exosome from the Peripheral Blood

The Exosome Isolation Q3 kit for serum (Wayen Biotechnologies, Shanghai, China) was applied for exosome isolation according to the manufacturer’s instructions as previously described [25]. Briefly, the serum was centrifuged at 3000× *g* for 10 min at 4 °C. In addition, 200 μL pretreated serum was mixed with 50 μL Reagent A. The mixture was incubated at 4 °C for 30 min. The mixture was centrifuged at 3000× *g* for 10 min at room temperature. Then the supernatant was removed, the pellet at the bottom was centrifuged, and the residual supernatant was discarded. Subsequently, the pellet was resuspended in 200 μL 1 × PBS. Fifty μL Reagent B was added and mixed by pipetting up and down. The mixture was incubated at 4 °C for 30 min. Afterwards, the mixture was centrifuged at 3000× *g* for 10 min at room temperature to remove the supernatant. The pellet containing the sEVs was resuspended in 100 μL 1 × PBS, and then the suspension was filtrated by 0.22 μm membrane. The product was stored at −80 °C for later use. The particle size distribution and concentration of the sEVs were evaluated by nanoparticle tracking analysis (NTA) using the NanoSight NS 300 system (NanoSight Technology, Malvern, Worcestershire, UK) or using ZetaView S/N 17-310 (Particle Metrix, Meerbusch, Germany) with the ZetaView software 8.04.02. The morphology of sEVs was observed with Tecnai G2 Spirit BioTWIN transmission electron microscope (TEM; FEI Company, Hillsboro, OR, USA) under 80 kV acceleration voltage. Exosome markers, including Alix, CD9, and CD63, were detected by Western blotting.

### 2.3. Cell Culture and RNA Transfection

HT22 mouse hippocampal neuronal cells were obtained from the Cell Bank of the Chinese Academy of Science (Shanghai, China) and cultured in Dulbecco’s Modified Eagle’s Medium (DMEM) supplied with 10% fetal bovine serum (FBS; Gibco, Grand Island, NY, USA) and 2 ng/mL FGF-2 at 37 °C in a humidified incubator with 5% CO2. HT22 cells were seeded in a six-well plate (40 × 10^4^ per well) and incubated for 24 h. When cells reached 70%~80% confluence, the culture medium was replaced by fresh medium. Five μL (2 μg) mmu-miR-342-5p inhibitor, mmu-miR-342-5p mimics, or the corresponding normal control (GenePharma, Shanghai, China) and 5 μL Lipofectamine™ 2000 transfection reagent (Invitrogen, Carlsbad, CA, USA) were diluted in OPTI-MEM (Gibco, Grand Island, NY, USA), respectively. After incubating for 5 min under room temperature, 50 μL diluted transfection reagent and 50 μL diluted RNA oligomers were mixed (the final concentration of miRNA molecules to 4 μg/100 μL). After 20 min, the mixture was added to the medium. Cells were cultured for another 24 h for the subsequent experiments. Sequences of mmu-miR-342-5p mimics (double chains), mimics normal control (NC, double chains), mmu-miR-342-5p inhibitor, and inhibitor NC were AGGGGUGCUAUCUGUGAUUGAG, CUCAAUCACAGAUAGCACCCCU; UUCUCCGAACGUGUCACGUTT, ACGUGACACGUUCGGAGAATT; CUCAAUCACAGAUAGCACCCCU; and CAGUACUUUUGUGUAGUACAA, respectively.

### 2.4. RNA Extraction and Quantitative RT-PCR

Total RNA was extracted with Trizol extraction kit (Invitrogen, Carlsbad, CA, USA). RNA quantification and quality assurance were performed by NanoDrop ND-1000 (NanoDrop, Wilmington, DE, USA). RNA Integrity and gDNA contamination were tested by agarose gel electrophoresis. Reverse transcription was performed using RevertAid First Strand cDNA Synthesis Kit (K1622, Thermo Fisher Scientific, Waltham, MA, USA). A total of 1 μg RNA from each sample was reversely transcribed using random primer in accordance with the manufacturer’s instructions. Briefly, 1 μL RNA template and 1 μL random primer were added into RNase-free ddH2O to the final volume of 12 μL, which was mixed with 1 μL transcriptase, 1 μL Protector RNase Inhibitor, 2 μL dNTP mix, and 4 μL 5 × Buffer. Aliquots of cDNA were stored at −80 °C for subsequent use. SYBR-Green PCR Master Mix (Roche, Basel, Switzerland) containing 0.3 μL 10 μM forward (F) and 0.3 μL 10 μM reverse (R) primers were added to 384-well PCR plate. In addition, 1 μL cDNA was pipetted in the wells. PCR was performed by ABI QuantStudio 6 Flex Real-Time PCR system (Applied Biosystems, Foster City, CA, USA). Melting curve analysis was used to monitor the product specificity. U6 snRNA or β-actin was used as inner control for normalization. The levels of BACE1 and miR-342-5p were calculated with 2^−ΔΔCt^ method. The primers used in this study are listed in Table 2.

### 2.5. Western Blotting

Exosomes or sEVs were lysed in the isopyknic lysis buffer containing 50 mM Tris (pH 7.4), 150 mM NaCl, 1% TritonX-100, 1% sodium deoxycholate, 0.1% SDS, and 1 mM Phenylmethanesulfonyl fluoride (PMSF) on ice for 10 min. The samples were then boiled for 10 min after adding loading buffer, and the protein concentrations were determined by BCA method. Equal amounts (10 μg) of proteins underwent SDS-PAGE and were electroblotted onto PVDF membranes. The membranes were then blocked with 5% nonfat milk in PBS-Tween 20 (PBST; 0.05%) for 1 h and incubated with primary antibody (1:1000 in PBST) at 4 °C overnight. Antibody for APP (#2452S) was obtained from Cell Signaling Technology (CST, Danvers, MA, USA); antibodies for Aβ42 (Ab201060) and BACE1 (Ab108394) were from Abcam (Abcam, Cambridge, MA, USA). After being washed three times in PBST (0.1% Triton X-100 in PBS), the membranes were incubated with appropriate secondary antibodies (1:5000) at room temperature for 1 h. After washing, the immunoreactive bands were visualized by 3,3′-diaminobenzidine (DAB). The integrated densities were analyzed by Image J software (NIH, Bethesda, MD, USA).

### 2.6. Dual-Luciferase Reporter Assay

To explore the mechanisms of miR-342-5p-modulated amyloid pathology, we predicted its potential target genes using TargetScan (http://www.targetscan.org/, accessed on 21 September 2020). The candidate target genes concerning neural function were listed in Table 3. For vector cloning, the mouse full-length *Bace1* 3′-UTR was synthesized with cleavage sites of *XhoI* and *NotI* on ABI3900 DNA synthesizer (Applied Biosystems, Foster City, CA, USA), and the predicted miR-342-5p target site in the *Bace1* 3′-UTR constructs was mutated using the QuikChange Lightning Site-directed Mutagenesis Kit (Agilent Technologies, Santa Clara, CA, USA). The obtained flat-end sequences were cloned into PUC57 vector. The constructs were digested with *XhoI* and *NotI* at 37 °C for 2 h, separated by 1% agarose gel electrophoresis, and retrieved with QIAquick Gel Extraction Kit (QIAGEN, Valencia, CA, USA). The psiCHECK2 dual-luciferase reporter vectors (Promega, Madison, WI, USA) were digested with *XhoI* and *NotI* overnight at 37 °C and separated by 1% agarose gel to retrieve the linear products. The target DNA segment was cloned into psiCHECK2 vector by T4 DNA ligase overnight at 16 °C and transformed into E. coli DH5α cells. The positive clone was selected by LB medium with ampicillin or kanamycin. *XhoI*/*NotI*-blunt fragment was subjected to electrophoresis and sequencing.

Before transfection, HEK 293T cells at a density of 5 × 10^4^ cells/well were seeded in 48-well plates in RPMI 1640 medium without antibiotics. When 70~80% confluence reached, cells were transfected with wild-type or mutant psiCHECK-BACE1 plasmid as well as mmu-miR-342-5p mimics or the miR negative control (miR-NC) by Lipofectamine 2000 (Invitrogen, Carlsbad, CA, USA). After 6 h, the medium with plasmids was substituted with fresh medium, and cells were further cultured for 48 h. Then the cells were lysed with lysis buffer. Luciferase activity was measured with the Dual-Luciferase Reporter Assay System (Promega, Madison, WI, USA), and the firefly luciferase activity was used as an internal control. All tests were performed in triplicate.

### 2.7. Mouse Hippocampus Exosome Isolation and MiR Mimics Transfection

Four six-month-old male APP mice and four age- and gender-matched C57BL/6 wild-type counterparts were maintained under pathogen-free conditions on a 12 h light–dark cycle with continuous access to food and water. When the mice reached 12 months of age, they were executed to collect the brains. SEVs were isolated from the brains of APP and C57BL/6 mice as previously described [26]. Briefly, the brains were dissected to isolate the hippocampi which were then cut into small pieces. Tissue was incubated in serum-free Hibernate A (A12475-01, Life Technologies) with 0.2% *w/v* collagenase type Ⅲ at 37 °C for 30 min. Protease inhibitors with 50 mM NaF and 200 nM Na_3_VO_4_ in ice-cold Hibernate-A were applied to cease the reaction. Tissues were transferred into a 10 mL plastic pipette and loosened by gently pipetting the tissue pieces up and down. The solution was subjected to centrifugation at 4 °C, 300× *g* for 10 min; 2000× *g* for 10 min, and 10,000× *g* for 30 min to discard cell pellets, membrane fragments, and other debris. The acquired supernatant was subjected to membrane filtration (0.22 μm) and centrifugation at 120,000× *g* for 120 min. The precipitated sEVs were rinsed twice and resuspended in PBS for use or storage at −20 °C. Cy5-labeled miR-342-5p mimics or mimics NC was transfected into sEVs with Exo-Fect Transfection Kit (#EXFT10A-1, SBI, Palo Alto, CA, USA) according to the manufacturer’s protocol. 

### 2.8. Incubation of HT22 Cells with Dil-Labeled sEVs

The sEVs (1 µg/µL of protein concentration) were stained with 1 mM fluorescent dye Dil (1,1′-dioctadecyl-3,3,3′,3′-tetramethylindocarbocyanine perchlorate; Thermo Fisher Scientific, Waltham, MA, USA) at a volume ratio of 500:1 at 37 °C for 1 h and then Dil-labeled sEVs were washed three times with PBS before incubation. HT22 cells were incubated with Dil-labeled sEVs (final concentration: 10 μg/mL) at 37 °C for 24 h. Then the culture medium was removed, and cells were washed three times with serum-free medium to remove the surface-bound sEVs. Nuclei were stained with fluorescent dye DAPI (4′, 6-diamidino-2-phenylindole). Images were taken with a fluorescence microscope (Nikon, Tokyo, Japan).

### 2.9. Statistical Analysis

SPSS 19.0 (SPSS, Chicago, IL, USA) and GraphPad Prism 5.0 (GraphPad Software, GraphPad, San Diego, CA, USA) were used for the statistical analysis. Normally distributed data after the normality test are presented as mean ± standard deviation. Non-normally distributed data are presented using the median and quartile. Independent t-test or nonparametric test was used to compare the differences between two groups. Analysis of variance or nonparametric test was used to compare the differences in multiple groups. Receiver operating characteristic (ROC) curve analysis was applied to evaluate the performance of the diagnostic accuracy. *p*-value < 0.05 was considered statistically significant.

## 3. Results

### 3.1. MiR-342-5p was Dysregulated in Serum sEVs from AD Patients

Circulating sEVs were purified from AD patients and healthy controls (HC). Nanoparticle tracking analysis (NTA) was performed to analyze the concentration and particle size distribution of the sEVs. The purified sEVs from AD patients (Figure 1a) and HCs (Figure 1b) presented a single sharp peak at approximately 100~150 nm in diameter. The presence of exosomal protein markers CD9, CD63, and Alix (PDCD6IP) was confirmed by Western blot assay. As shown in Figure 1c, the sEVs were positive for the exosome markers CD9, CD63, and Alix. Then the sEVs were observed using the transmission electron microscope (TEM), and the sEVs showed a typical cup-shaped morphology (Figure 1d). These data demonstrated that the vesicles isolated from the human sera were sEVs based on their size, morphology, and expressions of exosomal markers. Subsequently, the levels of miR-342-5p in circulating sEVs were quantified by RT-PCR assay. MiR-342-5p was significantly downregulated in serum sEVs from AD patients compared with those from the HC group (Figure 1e, *p* = 0.0291), implying the potential of miR-342-5p in AD diagnosis and clinical treatment. Considering the lack of universal loading control for exosomal miRNA analysis, some internal RNAs like miR-191-5p, miR-16, and U6 snRNA were widely used as controls for the analysis of exosomal noncoding RNAs [27]. Additionally, miR-425-3p was used as an invariant miRNA for statistical analysis of plasma exosomal miRNA according to previous work [28]. We then normalized the data of miR-342-5p for comparison by miR-191-5p and miR-425-3p, respectively, and analyzed the variation trend of the relative level of miR-342-5p. The data showed that the ratios of miR-342-5p to miR-191-5p in sEVs presented a significant difference between the two groups (Figure 1f; from 16 AD patients and 16 healthy controls; *p* = 0.0027). Meanwhile, no significant difference was observed in miR-425-3p levels in sEVs between the AD and HC groups (Figure 1g; from 16 AD patients and 16 HC; *p* = 0.7774). In addition, there were no significant changes in the ratio of miR-342-5p to miR-425-3p in sEVs from the AD group compared to those from healthy controls (Figure 1h; from 16 AD and 16 HC; *p* = 0.5588). To estimate the diagnostic performance of circulating sEV miR-342-5p, the receiver operating characteristic (ROC) curve was created to discriminate AD patients from healthy controls. The ROC analysis highlighted a diagnostic role for miR-342-5p with an area under the curve (AUC) of 0.643 (Figure 1i).

### 3.2. MiR-342-5p Targets Bace1 in Mouse Hippocampal HT22 Neurons

The target genes of mmu-miR-342-5p were predicted by the bioinformatics method. TargetScan7.1, a sequence-based tool, retrieved 4127 predicted mRNA targets with a cumulative weighted context++ score and total context++ score. Table 3 lists the predicted target genes of mmu-miR-342-5p concerning neural function, and data show that *Bace1* might be a potential target gene of mmu-miR-342-5p. Then, mmu-miR-342-5p mimics, mmu-miR-342-5p inhibitor, or their corresponding negative control (NC) was transfected into HT22 cells. Mmu-miR-342-5p level was effectively modulated by mmu-miR-342-5p mimics and mmu-miR-342-5p inhibitor (Figure 2a; *, *p* < 0.05, ***, *p* < 0.001). Furthermore, RT-PCR was performed to evaluate *Bace1* mRNA levels in HT22 cells. Mmu-miR-342-5p inhibitor significantly elevated the relative level of *Bace1* mRNA while the mimics lowered the level of *Bace1* mRNA in contrast (Figure 2b; *, *p* < 0.05). The proteins of APP, BACE1, and Aβ42 were visualized by Western blotting (Figure 2c). Our data revealed that the levels of APP, BACE1, and Aβ42 were obviously elevated by mmu-miR-342-5p inhibitor compared with inhibitor-NC (Figure 2d; *, *p* < 0.05, **, *p* < 0.01) whilst the mimics significantly lowered the protein levels of APP, BACE1, and Aβ42 (Figure 2e; *, *p* < 0.05). Subsequently, we performed a dual-luciferase reporter assay to test the interaction of *Bace1* and miR-342-5p. Wild-type 3′-UTR sequence in *Bace1*, the predicted target sequence of miR-342-5p, was cloned into the psiCHECK2 dual-luciferase reporter vector. The *Bace1* mutant sequence which presents partial complementary binding with miR-342-5p was designed and substituted for the wild-type *Bace1* 3′-UTR in the vector (Figure 2f). Luciferase activity in HEK293T cells transfected with the wild-type *Bace1* vector and miR mimics was obviously lowered compared with the mimics NC. In contrast, for the *Bace1* mutant vector, no obvious change in the luciferase activity was found even when treated with miR-342-5p mimics (Figure 2g; ***, *p* < 0.001).

### 3.3. MiR-342-5p-Dysregulated sEVs from Hippocampus of APP Mouse Exacerbate Aβ42 Formation in HT22 Neurons

SEVs were isolated from the hippocampus regions of the brains of 12-month-old male transgenic mice (Figure 3a–d) harboring the Swedish mutant form of human APP (Tg2576) with age- and gender-matched nontransgenic C57BL/6 mice of the same genetic background as the control (Figure 3e–h). Nanoparticle tracking analysis (NTA) was performed to evaluate the concentration and particle size distribution of the sEVs from APP mice (Exo-APP, Figure 3a,b) and C57BL/6 littermates (Exo-CTL, Figure 3e,f). The particle size of Exo-APP and Exo-CTL was 100-150 nm in diameter, and the exosome volume peaks at approximately 100-200 nm in diameter. In Figure 3c,g, NTA video visualized sEVs by light scattering, revealing the presence of vesicles of different sizes. Exo-APP (Figure 3d) and Exo-CTL (Figure 3h) were visualized by transmission electron microscopy (TEM). They presented a typical exosomal cup-shaped morphology. The levels of CD9, CD63, and GAPDH were detected by Western blot analysis. The isolated vesicles, Exo-APP (APP-EXO), and Exo-CTL (C57-EXO) were positive for the exosomal markers CD9 and CD63 but negative for GAPDH, implying scarce contamination with cellular contents (Figure 3i). Furthermore, the levels of miR-342-5p in both Exo-APP and Exo-CTL were detected using RT-PCR, and the concentration of miR-342-5p was significantly lower in Exo-APP compared with that in the Exo-CTL group (Figure 3j; **, *p* < 0.01). Then, the mRNA levels of BACE1 in Exo-APP and Exo-CTL were also quantified by RT-PCR assay, and the data showed that the BACE1 mRNA level was significantly higher in Exo-APP compared with the control group (Figure 3k; **, *p* < 0.01). To determine whether HT22 neurons effectively take up the hippocampal sEVs, the purified sEVs were labeled with the fluorescent dye Dil (red) and incubated with HT22 cells for 24 h. After the cells were washed three times with a serum-free medium to eliminate the surface-bound sEVs, the cell nuclei were stained with the fluorescent dye DAPI (blue), and cells were visualized with a fluorescence microscope. The fluorescent images demonstrated that Dil-labeled sEVs (red small dots) were localized around the cell nuclei (blue-filled ellipses), revealing their efficient incorporation into the HT22 cells. The scale bar at the lower right corner of each image indicates 50 μm in length (Figure 3l). After HT22, cells were incubated with Exo-APP or Exo-CTL, the mRNA levels of BACE1 in the cells were quantified with RT-PCR. The data showed that Exo-APP obviously elevated BACE1 mRNA level in HT22 cells more than Exo-CTL (Figure 3m; ***, *p* < 0.001). Western blot assay of the HT22 cell lysates revealed that the protein levels of APP, BACE1, and Aβ42 presented an obviously lower level in HT22 cells treated with Exo-CTL, instead of those with Exo-APP (Figure 3n). The protein levels in HT22 cells were quantified using densitometry and normalized to the inner control GAPDH. The relative protein levels were summarized in a bar chart depicting that APP, BACE1, and Aβ42 were obviously elevated by Exo-APP compared with Exo-CTL (Figure 3o; *, *p* < 0.05, **, *p* < 0.01, ***, *p* < 0.001).

### 3.4. MiR-342-5p Enrichment in Hippocampal sEVs Rescued Aβ42 Formation in Recipient HT22 Neurons

To further evaluate the role of miR-342-5p in Aβ pathology, sEVs from hippocampi of the transgenic mice brains were treated with miR-342-5p mimics or its negative control (NC), respectively. The fluorescent microscope images showed that the Dil-labeled sEVs (small red dots) appeared around the nuclei (blue ellipses) of HT22 cells, verifying the uptake of Dil-labeled sEVs into living HT22 cells. Scale bar = 50 μm (Figure 4a). RT-PCR data demonstrated that the level of miR-342-5p in the treated sEVs was profoundly elevated by miR-342-5p mimics compared with the mimics NC (Figure 4b; **, *p* < 0.01). After being incubated with the control Exo-APP (EXO-miR-342-5p-NC) or miR-342-5p-enriched Exo-APP (EXO-miR-342-5p-mimic), the protein levels of APP, BACE1, and Aβ42 were evaluated using Western blot assay. GAPDH was applied as an inner control (Figure 4c). The protein levels in HT22 cells were quantified using densitometry and normalized to the inner control GAPDH. The data revealed that HT22 cells that were treated with EXO-miR-342-5p-mimic presented an obvious decrease in the protein levels of APP, BACE1, and Aβ42 compared with Exo-APP per se (Figure 4d; *, *p* < 0.05, **, *p* < 0.01), highlighting a key role for sEVs derived miR-342-5p which mediated BACE1 deactivation and ameliorated Aβ42 formation in the recipient neurons.

## 4. Discussion

Dysregulation of miRNAs is involved in the pathogenesis of neurological diseases. Our previous work demonstrated that a panel of miRNAs in circulating sEVs may serve as a biomarker for AD diagnosis [25]. Lately, increasing evidence suggests a pivotal role for miR-342-5p in neurodegeneration. Noteworthily, a recent study revealed that AD patients with lower miR-342-5p levels in peripheral blood were prone to suffering from more severe cognitive decline [29]. However, its underlying mechanisms in the severity of AD remain ambiguous. Thus, the present work was designed to evaluate the potential for sEVs miR-342-5p in amyloid pathology in cultured neurons. Our data demonstrated that AD patients possessed a lower level of miR-342-5p in sEVs in the peripheral blood, and sEV-derived miR-342-5p ameliorates Aβ formation in the recipient neurons by modulating BACE1 expression.

SEVs (or exosomes) originate within multivesicular bodies (MVBs). Endosome membrane invagination allows the sequestering of the cytosolic cargos to form exosomes, which are released by membrane fusion of MVB with the cell membrane [30]. Exosomes mediate cellular communication via transferring biological molecules, such as protein, mRNA, and miRNA, to recipient cells through direct membrane fusion, endocytosis, or receptor–ligand interaction. The signaling pathways inside the recipient cells are then accurately modulated, and their pathophysiological reactions are accordingly altered. Exosomes are continuously released and absorbed by cells. While the released exosomes are incorporated into recipient cells through endocytosis, they return to MVBs for recycling [31]. In a sense, the contents of the exosomes that are incorporated into target cells determine the cellular behavior accordingly. Intriguingly, miRNAs and their repressible mRNAs are enriched in endosomes and MVBs [32], implying that endosomes and MVBs are the sites of miRNA target recognition and subsequent post-translational gene silencing. The interaction of miRNA and mRNA may be modulated in mammalian cells. In the presence of target mRNA, miRNAs are prone to delivery by MVB-secreted exosomes and this target mRNA-driven miRNA export process was partially retarded by Argonaute 2 (Ago2)-interacting protein GW182B [33].

Considering the good stability, bioactivity, target specificity, and availability to cross the blood–brain barrier, there has been increasing interest in engineering exosomes as delivery vehicles of therapeutics. Previous work demonstrated that modified exosomes (targeting brain neurons via rabies virus glycoprotein (RVG) peptide on the exosomal membrane) that were loaded with exogenous siRNA against BACE1 were intravenously injected in C57BL/6 mice, leading to a specific decrease in the mRNA and protein levels of BACE1 in the cortical neuron cells of the mice [34]. Furthermore, RVG-modified exosomes enriched with miR-21 inhibitors effectively induced the expression of target genes PTEN and PDCD4 in recipient cells and reduced the brain tumor size [35]. These studies as well as our work implied that modified exosomes capable of delivering RNA oligonucleotides into targeted cells represent a promising strategy for AD clinical therapy. Compared to the siRNA-based strategy, miRNA is an endogenous and naturally occurring noncoding RNA involved in regulating multiple gene expressions. Thus, exosomal delivery of miRNAs might provide a better approach for gene modulation. In addition, exosomal miRNA-mediated signal transduction in the recipient cells needs further evaluation.

Exosomes that are internalized into the recipient cells will be transported into endosomes [36]. APP processing mainly occurs in early endosomes. The soluble Aβ40 and Aβ42 predominantly colocalize with the early endosome marker Rab5 in neurons. Before a clinical diagnosis of AD, a specific feature of irreversible endosome swelling in neurons may appear and predicts the earliest onset of AD [37]. In AD patient brains and mutant human APP transgenic mice, BACE1 accumulates within late endosomes at the synapses. Late endocytic distribution of BACE1 increased APP processing at presynaptic sites [38]. Aβ occurs predominantly in endosomes, and C-terminal fragments (CTFs) of APP, and α-, and β-secretases have been identified in exosomes [39], indicating that cleavage of APP occurred in the vesicles might be regulated by the exosomal cargos.

Aβ originates from a transmembrane glycoprotein APP depending on the sequential cleavage by BACE1 at the N-terminus and γ-secretase at the C-terminus. It consists of 36 to 43 amino acid residues, and Aβ42 occurs most commonly in the plaques of AD brains [40]. Aβ maintains metastable α-helix confirmation in lipid environment, and Aβ monomers are prone to aggregate into oligomers or fibers with the conformation changing into a misfolded β-sheet [41,42,43]. In the last few decades, numerous researchers have been focusing on antagonizing the accumulation of Aβ in the AD brain. Unfortunately, drugs directly targeting Aβ production, clearance, and aggregation showed few improvements in clinical symptoms. Moreover, the inhibitors for BACE1 and γ-secretase may worsen cognitive losses and psychiatric and clinical conditions in AD patients [44]. Critically, BACE1 is prone to binding with non-amyloid substrates instead of APP whereas endosomally targeted BACE1 inhibitors specifically blocked the APP cleavage in many cell systems but not non-amyloid substrates. Non-amyloid substrates were processed independently of endocytosis [45]. These data may imply that APP processing and Aβ formation occur in early endosomes depending on the endocytosis of circulating exosomes. As illustrated in previous work, BACE1 distribution into early endosomes was facilitated by small GTPase ADP ribosylation factor 6 (ARF6). Modulation of the ARF6 level or its activity led to aberrant BACE1 sorting, ultimately altering APP processing and Aβ formation [46]. Furthermore, faster trafficking of BACE1 from early endosome to recycling endosome also resulted in reduced APP processing and Aβ production [47]. These data also highlighted the critical role of exosome–endosome recycling in AD.

Our findings revealed that sEV-transferred miR-342-5p may ameliorate amyloid formation in the recipient neuron cells, implying that normal exosome release of functional miRNA(s) targeting BACE1 plays a key role in AD pathology. Considering that BACE1 participates in APP processing to form amyloid proteins in early endosomes, exosomal transfer of miRNA(s) for the modulation of BACE1 expression is practically feasible. However, the exosomes or sEVs in human circulating blood can be engulfed by different cells or tissues. Although the eminent work provided a tool for the exosomal delivery of siRNAs to neurons, microglia, and oligodendrocytes in the brain [34], its targeting specificity appears to be unpredictable and needs further improvement.

## 5. Conclusions

In the present study, we demonstrated the miR-342-5p level in serum sEVs in AD patients was obviously reduced compared with healthy controls. MiR-342-5p facilitated the degradation of BACE1 mRNA in the cells by targeting its 3′-UTR sequence. SEVs from APP mice (Exo-APP) which contain lower miR-342-5p than those from C57BL/6 mice promoted Aβ production. Crucially, the modified miR-342-5p-enriched Exo-APP ameliorated amyloid pathology in the recipient neuron cells compared with the naïve Exo-APP. Our work might provide a clue for dementia-relevant BACE1 modulation through exosome-mediated delivery of miRNAs to the endosomal system in the target neuron cells.

## Figures and Tables

**Figure 1 cells-11-03830-f001:**
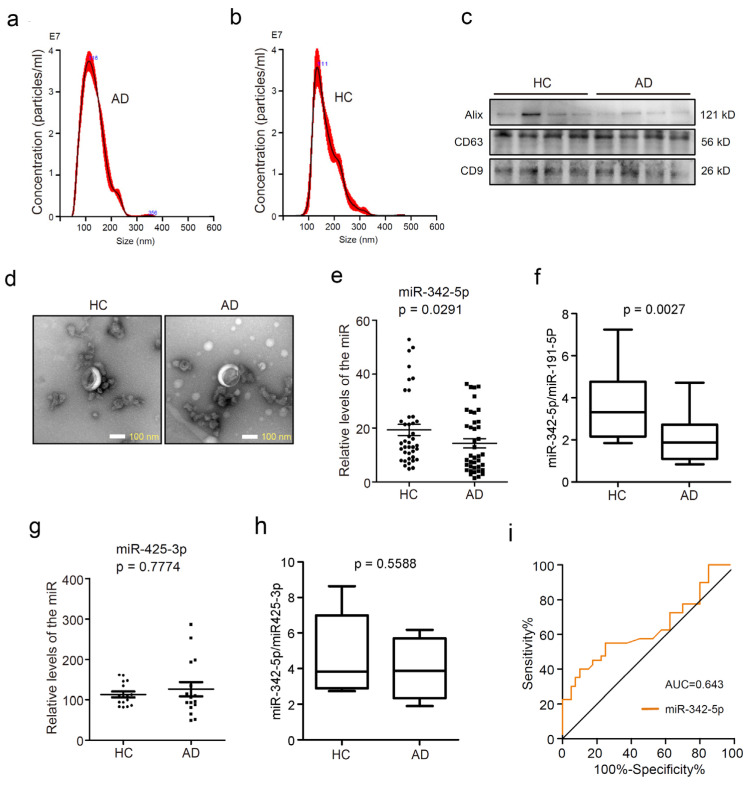
Circulating sEV miRNA-342-5p was downregulated in patients with Alzheimer’s disease (AD) compared with healthy controls (HC). (**a**) Nanoparticle Tracking Analysis (NTA) histograms of purified circulating sEVs from AD patients. The horizontal axis represents the particle diameters (nm), and the longitudinal axis depicts particle concentrations (particles/mL) at each diameter. (**b**) NTA histograms of purified circulating sEVs from healthy controls; (**c**) tetraspanins CD63 and CD9, and Alix, the exosomal biomarkers were detected using Western blot assay. (**d**) The transmission electron microscopy (TEM) images exhibited the morphology of sEVs from healthy controls (left) and AD patients (right). Scale bars represent 100 nm. (**e**) RT-PCR assay was performed to quantify the level of miR-342-5p in circulating sEVs from healthy controls or AD patients, *p* = 0.0291. (**f**) The ratios of miR-342-5p to miR-191-5p in sEVs from 16 AD patients and 16 healthy controls were calculated, *p* = 0.0027. (**g**) MiR-425-3p levels in sEVs were evaluated in the HC and AD groups, *p* = 0.7774. (**h**) The ratios of miR-342-5P to miR-425-3p were calculated, *p* = 0.5588. (**i**) The receiver operating characteristic (ROC) curve and area under the curve (AUC) were applied to estimate the diagnostic performance of sEVs miR-342-5p in distinguishing AD patients from normal individuals.

**Figure 2 cells-11-03830-f002:**
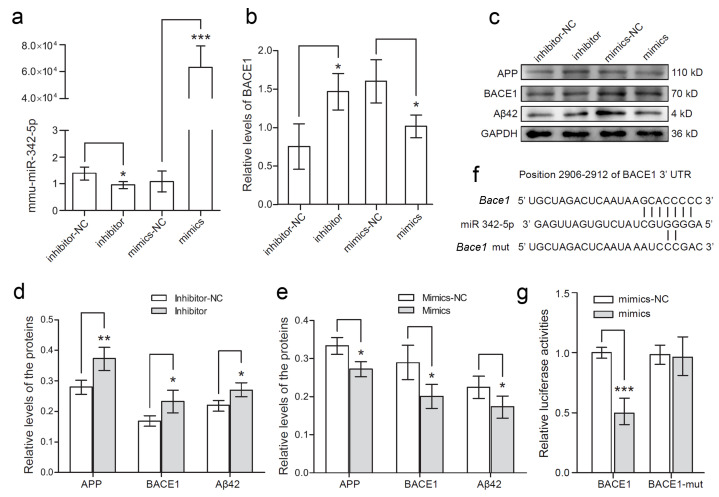
MiRNA-342-5p modulated the expression of *Bace1* by targeting its 3′-UTR in mouse hippocampal HT-22 neurons. (**a**) MiR-342-5p was detected by RT-PCR assay after the cells were transfected with miR-342-5p inhibitors or mimics or their corresponding negative control (NC). *, *p* < 0.05, ***, *p* < 0.001. (**b**) BACE1 was detected by RT-PCR assay after the cells were transfected with the above RNA oligonucleotides. *, *p* < 0.05. (**c**) APP, BACE1, and Aβ42 were detected by Western blot assay. GAPDH was used as the internal control. The experiments were performed in triplicate. (**d**) Relative quantification of APP, BACE1, and Aβ42 immunoblot bands was performed by densitometry normalized to GAPDH after the cells were transfected with miR inhibitors and its negative control (NC). *, *p* < 0.05, **, *p* < 0.01. (**e**) Relative quantification of APP, BACE1, and Aβ42 immunoblot bands was performed by densitometry after the cells were transfected with miR mimics and the negative control (NC). *, *p* < 0.05. (**f**) A sequence alignment was presented for miR-342-5p with predicted sequence from wild-type *Bace1* and its mutant (mut) sequence. (**g**) Relative luciferase activities of wild-type and mutant *Bace1* reporter were measured in HEK293T cells overexpressing miR-342-5p mimics or the negative control (NC). ***, *p* < 0.001.

**Figure 3 cells-11-03830-f003:**
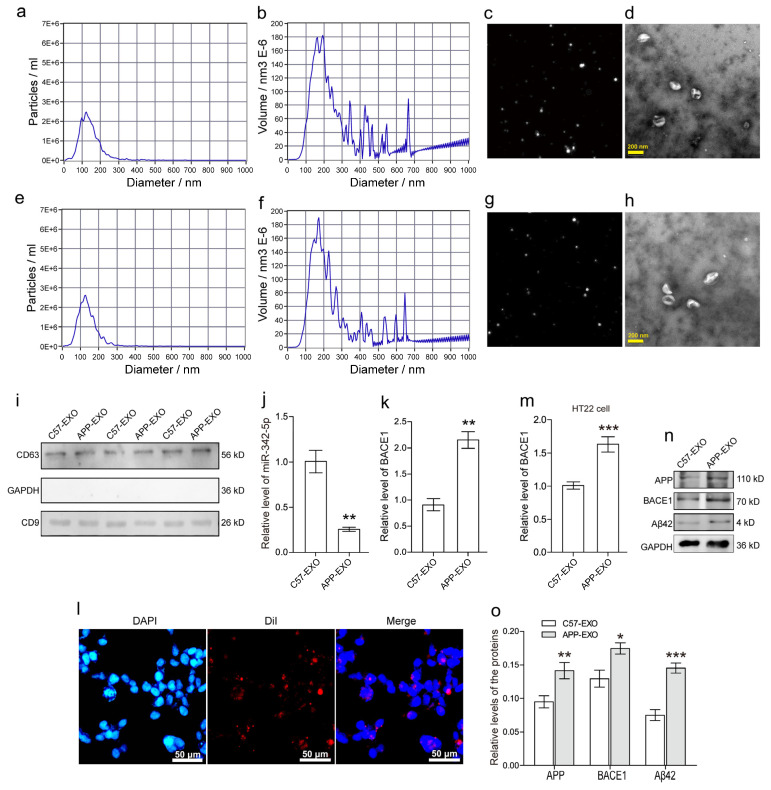
SEVs from APP transgenic mice distinctly promoted Aβ formation in mouse hippocampal HT-22 cells. SEVs were purified from APP mice (**a**–**d**) and C57BL/6 control mice (**e**–**h**). The sEVs were subjected to nanoparticle tracking analysis with a ZetaView instrument utilizing the properties of light scattering and Brownian motion to evaluate the exosome size distribution. In (**a**,**e**), the horizontal axis shows the particle diameters (nm) in the samples, and the vertical axis displays the concentrations (particles/mL) at a certain size. In (**b**,**f**), the horizontal axis shows the particle diameters (nm), and the vertical axis displays the particle volumes (×10^6^ nm^3^) at a certain diameter size. In (**c**,**g**), NTA video visualized light scattering sEVs, revealing the presence of vesicles of different sizes. In (**d**,**h**), the morphology of sEVs was visualized by transmission electron microscope (TEM). The scale bar is 200 nm. In (**i**), exosome markers CD9 and CD63 were detected by Western blot assay. GAPDH was used as a negative control. In (**j**), the RT-PCR assay quantified the relative levels of miR-342-5p in sEVs from APP and C57BL/6 mice. **, *p* < 0.01. In (**k**), RT-PCR quantified the relative levels of BACE1 in APP and C57BL/6 sEVs. **, *p* < 0.01. In (**l**), a Nikon fluorescent microscopy was used to visualize the internalization of Dil-labeled sEVs into HT22 cells. Cell nuclei were detected with the fluorescent dye DAPI. Scale bars represent 50 μm. In (**m**), an RT-PCR was applied to detect the relative levels of BACE1 in HT22 cells incubated with the sEVs from APP and C57BL/6 mice, respectively. ***, *p* < 0.001. In (**n**), a Western blot assay was used to detect APP, BACE1, and Aβ42 proteins in HT22 cells incubated with the sEVs from APP and C57BL/6 mice, respectively. The experiments were performed in triplicate. In (**o**), the intensities of the immunobands for APP, BACE1, and Aβ42, normalized to the inner control GAPDH were quantified by densitometry in HT22 cells incubated with sEVs from APP and C57BL/6 mice, respectively. *, *p* < 0.05, **, *p* < 0.01, ***, *p* < 0.001.

**Figure 4 cells-11-03830-f004:**
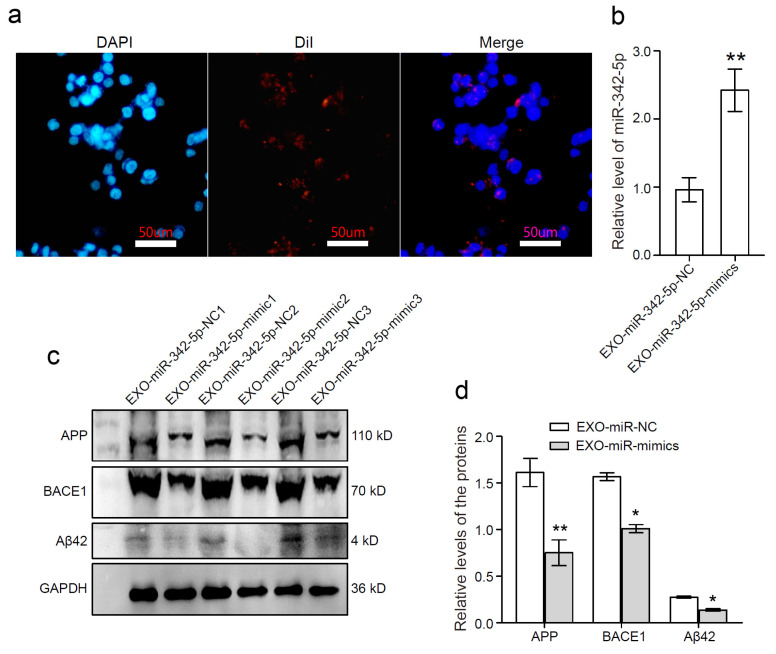
Enrichment of miR-342-5p ameliorated Aβ pathology evoked by Exo-APP in mouse hippocampal HT-22 neurons. (**a**) The internalization of Dil-labeled sEVs into HT22 cells was visualized using a Nikon fluorescent microscope. DAPI staining was used to detect cell nuclei. Scale bars represent 50 μm. (**b**) Exo-APP were transfected with miR-342-5p mimics (EXO-miR-mimics) or its negative control (EXO-miR-NC), respectively. The levels of miR-342-5p in the sEVs were evaluated using qPCR. **, *p* < 0.01. (**c**) HT22 cells were treated with the sEVs enriched with miR-342-5p mimics (EXO-miR-mimics) or its negative control (EXO-miR-NC). APP, BACE1, and Aβ42 proteins in HT22 cells incubated with EXO-miR-mimics or EXO-miR-NC were detected using Western blot assay. The experiments were performed in triplicate. (**d**) Relative levels of APP, BACE1, and Aβ42 in HT22 cells were quantified by densitometry. GAPDH was used as an inner control. *, *p* < 0.05, **, *p* < 0.01.

**Table 1 cells-11-03830-t001:** Demographic characteristics of the subjects enrolled in this study.

Characteristics	n (m/f)	Mean Age	MMSE Score
HC	40 (20/20)	77.1 ± 6.2	n.d.
AD	40 (20/20)	75.1 ± 7.5	6.55 ± 2.86

Note: Data are shown as mean ± SD. AD = Alzheimer’s disease; HC = healthy controls; n = number of subjects; m = male; f = female; MMSE = minimental state examination; n.d. = not done.

**Table 2 cells-11-03830-t002:** Primers for qPCR assay in this study.

Genes	Primer Sequences (5′ to 3′)
hsa-miR-342-5p	AGGGGTGCTATCTGTGAAAAA
hsa-U6	TTCGTGAAGCGTTCCATATTTT
mmu-GAPDH -F	CAAAATGGTGAAGGTCGGTGT
mmu-GAPDH -R	GAGGTCAATGAAGGGGTCGTT
mmu-miR-342-5p-F	CGCAGAGGGGTGCTATCTGT
mmu-miR-342-5p-R	AGTGCGTGTCGTGGAGTCG
mmu-U6-F	CGATACAGAGAAGATTAGCATGGC
mmu-U6-R	AACGCTTCACGAATTTGCGT
mmu-BACE1-F	GACCACTCGCTATACACGGG
mmu-BACE1-R	CTTCTCCGTCTCCTTGCAGT

Note: hsa, Homo sapiens; mmu, Mus musculus; miR = microRNA; F = forward primer; R = reverse primer; U6, U6 small nuclear RNA; GAPDH, Glyceraldehyde 3-phosphate dehydrogenase; BACE1, beta-site APP-cleaving enzyme 1.

**Table 3 cells-11-03830-t003:** Predicted target genes of miR-342-5p concerning neural function.

Ortholog of Target Gene	Representative Transcript	Gene Name	Cumulative Weighted Context++ Score	Total Context++ Score
*Nsmf*	ENSMUST00000100334.5	NMDA receptor synaptonuclear signaling and neuronal migration factor	−0.43	−0.43
*Smpd3*	ENSMUST00000067512.7	sphingomyelin phosphodiesterase 3, neutral	−0.02	−0.04
*Bace1*	ENSMUST00000034591.5	beta-site APP cleaving enzyme 1	−0.2	−0.38
*Nptxr*	ENSMUST00000175858.3	neuronal pentraxin receptor	−0.25	−0.65
*Syp*	ENSMUST00000069520.5	synaptophysin	−0.11	−0.11
*Syngap1*	ENSMUST00000081285.4	synaptic Ras GTPase activating protein 1 homolog (rat)	−0.62	−0.62
*Stxbp1*	ENSMUST00000077458.4	syntaxin binding protein 1	−0.37	−0.38
*Stx1b*	ENSMUST00000106267.3	syntaxin 1B	−0.24	−0.24
*Nf2*	ENSMUST00000056290.7	neurofibromatosis 2	−0.1	−0.1
*Nptx1*	ENSMUST00000026670.4	neuronal pentraxin 1	−0.14	−0.14
*Nrxn2*	ENSMUST00000113462.2	neurexin II	−0.24	−0.24
*Ngb*	ENSMUST00000110176.1	neuroglobin	−0.21	−0.25

## Data Availability

The data presented in this study are available upon reasonable request from the corresponding author. The data are not publicly available due to privacy or ethical restrictions.

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
