# Peer review of "Circulating Small Extracellular Vesicle-Derived miR-342-5p Ameliorates Beta-Amyloid Formation via Targeting Beta-site APP Cleaving Enzyme 1 in Alzheimer’s Disease"

_cells, 2022, doi:10.3390/cells11233830_

Round 1

Reviewer 1 Report

In the present manuscript the authors have evaluated the levels of miR-342-5p in the serum exosomes of alzheimer's disease patients. This investigation is based on a previous report that miR-324-5p in the plasma of AD patients is low and associated with cognitive impairment.

The authors argue that miR-324 is contained in exosomes, is low in AD patients and targets BACE1. Ectopic expression of this miR by transfection of exosomes result in a reduction of BACE1 and a reduction in APP processing into Amyloid, therefore potentially improving AD pathogenesis.

Major considerations:

The notion that miR-324-5p is in exosomes is compelling but only partially supported by the results.

First, the authors have utilized bulk isolation methods for exosomes, therefore the term exosome here is incorrectly utilized. The authors should refer to the isolated vesicles as small EVs.

It is unclear why the authors chose two different methods for EV isolation: a precipitation reagent for the blood EVs and ultracentrifugation for mouse hippocampus derived EVs. The precipitation method is likely to isolate many contaminants from the serum, as seen in the TEM pictures (Fig 1d). The purity of the preparation can be evaluated by utilizing more markers in the western blot analysis such as markers for lipoproteins for examples (refer to MISEV guidelines published by the international society of extracellular vesicles).

The same technique should be used to argue that the mR is contained within the same type of vesicle.

Because of the limited characterization of the isolated vesicles, it is not possible to determine whether miR-342-5p is truly a cargo of the EVs or a co-isolate; the authors should consider performing experiment with nucleases and also to further purify the EVs, for example by density gradient or affinity methods.

Other minor points:

The efficiency of the transfection of EVs shown in fig 4b should be demonstrated: is the miR internal to the EVs? Consider performing RNAse treatment.

Amyloid beta 42 is not introduced in the discussion, this should be added for a broader audience

U6snRNA is not a proper normalization control for exosome/EV studies, in the lack of a proper control (that is stable and validated for this specific cohort of samples), a spike-in control may be more appropriate

In the western blotting section of the methods, please describe the composition of the lysis buffer and also add how the EVs were lysed and how much protein was loaded.

Author Response

Response to Reviewer 1 Comments

In the present manuscript the authors have evaluated the levels of miR-342-5p in the serum exosomes of Alzheimer's disease patients. This investigation is based on a previous report that miR-324-5p in the plasma of AD patients is low and associated with cognitive impairment.

The authors argue that miR-324 is contained in exosomes, is low in AD patients and targets BACE1. Ectopic expression of this miR by transfection of exosomes result in a reduction of BACE1 and a reduction in APP processing into Amyloid, therefore potentially improving AD pathogenesis.

Major considerations:

The notion that miR-324-5p is in exosomes is compelling but only partially supported by the results.

  1. First, the authors have utilized bulk isolation methods for exosomes, therefore the term exosome here is incorrectly utilized. The authors should refer to the isolated vesicles as small EVs.

Response 1: We really appreciate the reviewer’s comments. We described the isolated vesicles as small extracellular vesicles (sEVs) in the manuscript except for some references in which the phrase exosome had been utilized for their work.

  1. It is unclear why the authors chose two different methods for EV isolation: a precipitation reagent for the blood EVs and ultracentrifugation for mouse hippocampus derived EVs. The precipitation method is likely to isolate many contaminants from the serum, as seen in the TEM pictures (Fig 1d). The purity of the preparation can be evaluated by utilizing more markers in the western blot analysis such as markers for lipoproteins for examples (refer to MISEV guidelines published by the international society of extracellular vesicles).

Response 2: Thank you for the valuable comments. We used a precipitation reagent for the blood sEVs purification considering that the Exosome Isolation Q3 kit for serum from Wayen Biotechnologies presents a high purification rate of exosomes and a low protein contamination (e.g., high-abundant serum proteins). The influence of protein contamination could be reduced after the RNA isolation process for further small RNA research. Furthermore, a previous work (PMID: 32716349) compared the commercial reagents including Wayen and ultracentrifugation (UC), and demonstrated that “Wayen reagent seems to yield a transcriptome landscape more similar to UC than the other kits according to the clustering, venn diagraph and intra-kit correlation analysis”.

  1. The same technique should be used to argue that the miR is contained within the same type of vesicle. Because of the limited characterization of the isolated vesicles, it is not possible to determine whether miR-342-5p is truly a cargo of the EVs or a co-isolate; the authors should consider performing experiment with nucleases and also to further purify the EVs, for example by density gradient or affinity methods.

Response 3: We really appreciate the reviewer’s comments. To ascertain that miR-342-5p is contained in the purified small extracellular vesicles, we isolated sEVs from ten normal subjects using ultracentrifugation method. The same amount of Caenorhabditis elegans cel-39 miRNA was spiked into each sEVs sample as an external calibration. Then the samples were incubated with 5 μg/ml RNase A for 30 mins prior to RNA extraction. RT-PCR was performed to detect the CT values (total 45 cycles) for cel-mir-39 and hsa-mir-342-5p as shown in the following table. These data imply that miR-342-5p is contained in the isolated sEVs.

Sample Name

cel-mir-39

hsa-mir-342-5p

subject-1

7.966

26.917

subject-2

6.885

26.334

subject-3

7.359

22.897

subject-4

7.213

25.471

subject-5

7.655

24.373

subject-6

7.109

25.770

subject-7

7.355

23.268

subject-8

7.278

27.255

subject-9

7.770

26.498

subject-10

7.981

28.176

Other minor points:

  1. The efficiency of the transfection of EVs shown in fig 4b should be demonstrated: is the miR internal to the EVs? Consider performing RNase treatment.

Response 4: Thank you for the valuable comments. For the time limit, we just performed RNase A treatment for normal miR-342-5p and cel-miR-39 levels, if possible, we would continue our sEVs work with RNase A treatment.

  1. Amyloid beta 42 is not introduced in the discussion, this should be added for a broader audience.

Response 5: Thank you for the valuable comments. We further introduced amyloid beta 42 in the discussion section.

  1. U6 snRNA is not a proper normalization control for exosome/EV studies, in the lack of a proper control (that is stable and validated for this specific cohort of samples), a spike-in control may be more appropriate.

Response 6: We really appreciate the reviewer’s comments. Due to the lack of a proper control, some RNAs such as U6 snRNA, miR-191-5p and miR-16 have been utilized as internal controls during the analysis of exosomal non-coding RNAs (PMID30876419). We then analyzed the ratio of miR-342-5p/miR-191-5p and miR-342-5p/miR-425-3p in our data, and found that there was a significant difference in the ratio of miR-342-5p/miR-191-5p between the AD and healthy control groups (shown in Figure 1f and 1h). These data might to some extent reflect the variation trend for miR-342-5p in the circulating sEVs.

  1. In the western blotting section of the methods, please describe the composition of the lysis buffer and also add how the EVs were lysed and how much protein was loaded.

Response 7: We really appreciate the reviewer’s comments. We have described the composition of the lysis buffer and the way how sEVs were lysed, also the protein mounts for western blot. Exosomes or sEVs were lysed in the isopyknic lysis buffer containing 50 mM Tris (pH 7.4), 150 mM NaCl, 1% TritonX-100, 1% sodium deoxycholate, 0.1% SDS, and 1mM Phenylmethanesulfonyl fluoride (PMSF) for 10 min on ice. The samples were then boiled for 10 mins after adding loading buffer and the protein concentrations were determined by BCA method. Equal amounts (10 μg) of proteins underwent SDS-PAGE.

Reviewer 2 Report

This is an interesting study on the potential role of miR-342-5p exosomal trasfer in amyloid patology.

The exosomals miRNAs have been extensively studied in recent years and their dysregulation has been diplayed in the pathogenesis of AD and several other neurodegenerative diseases.  In addition, the exosomal miRNAs profile exhibits considerable potential as diagnostic and therapeuric tool for AD.

The paper is generally well written and structured, however, in my opinion, the Authors, stressing the fact that the exosomal trasfer of miR-342-5p ameliorates Abeta formation and amyloid pathology, could investigate also its effects on the main pathways affected in AD such as neuroinflammation and oxidative stress in Exo-APP HT-22 neurons.

The scale bars are not clearly visible in all the figures, for example in Fig.3

Author Response

Response to Reviewer 2 Comments

This is an interesting study on the potential role of miR-342-5p exosomal transfer in amyloid pathology.

The exosomal miRNAs have been extensively studied in recent years and their dysregulation has been displayed in the pathogenesis of AD and several other neurodegenerative diseases.  In addition, the exosomal miRNAs profile exhibits considerable potential as diagnostic and therapeutic tool for AD.

  1. The paper is generally well written and structured, however, in my opinion, the Authors, stressing the fact that the exosomal transfer of miR-342-5p ameliorates Abeta formation and amyloid pathology, could investigate also its effects on the main pathways affected in AD such as neuroinflammation and oxidative stress in Exo-APP HT-22 neurons.

Response 1: We really appreciate the reviewer’s suggestions. Neuroinflammation and oxidative stress play critical roles during AD pathogenesis. We would like to evaluate the effects of exosomal transfer of miR-342-5p on the pathways in our future work.

  1. The scale bars are not clearly visible in all the figures, for example in Fig.3

Response 2: Thank you for the valuable comments. We depicted the scale bars again to make them more clearly visible in figure 1, figure 3 and figure 4.

Reviewer 3 Report

The work entitled: Circulating Exosomal miR-342-5p Ameliorates Beta-Amyloid 2

Formation via Targeting Beta-Site APP Cleaving Enzyme 1 in 3 Alzheimer's Disease.

This work investigates the role of miR-342-5p in the biogenesis of Aβ and its value as a biomarker. They found that miR-342-5p decreases the expression of beta-site APP cleaving enzyme 1 (BACE1), Aβ42, and amyloid precursor protein (APP). Opposite results were found with the incubation of exosomes isolated from the hippocampus of a transgenic mouse model of AD compared to wild-type animals. Accordingly, those parameters were ameliorated when artificially loading exosomes with miR-342-5p mimic. In coherence with decreased miR-342-5p signaling during AD, serum exosomes from AD patients show decreased content of miR-342-5p compared to age-matched controls. This is valuable work; however, some improvements can still be performed.

Major comments,

The calculated fold change for serum exosomes was not normalized but compared to miR-425-3p levels. This is ok as there is no universal loading control for serum exosomes. However, it would be relevant to show how the data behave by normalizing each experiment with miR-425-3p; this could be included as supplementary material.

The table indicating the predicted targets of the miRNA must be somehow categorized. For instance, algorithms for prediction normally give a score for each gene. In addition, it is important to indicate whether the information is validated or not in the literature. Some other works may have validated the downregulation of BACE by miR-425-3p before, if this is the case, it must be indicated.

The number of replicates for the western blots in figure 2 must be indicated. In addition, the authors should clarify why the relative levels of protein are far from value 1 in the quantifications of control conditions (inhibitor NC and mimic NC). The same applies to figures 3O and 4d

To better show internalization, I would rather display the complete z stack of the confocal pictures. At least, I would show the upper, middle, and lower representative images.

The discussion has a lot of pitfalls. For instance, the statement “These data imply that APP processing and Aβ formation occur in early endosome depending on endocytosis of circulating exosomes” is not well-supported. Similarly, Figure 5, showing the Schematic figure depicting the possible mechanisms underlying the protective effect of exosomal miR-342-5p in Aβ pathology, has an erroneous interpretation of the cellular mechanisms of exosome endocytosis. I strongly suggest correcting the discussion and removing the schematic figure.

In addition, the discussion does not make completely clear what would be the translational impact of these results when considering the literature showing that decreasing BACE1 does not ameliorate the severity of the disease.

Author Response

Response to Reviewer 3 Comments

The work entitled: Circulating Exosomal miR-342-5p Ameliorates Beta-Amyloid 2 Formation via Targeting Beta-Site APP Cleaving Enzyme 1 in 3 Alzheimer's Disease.

 This work investigates the role of miR-342-5p in the biogenesis of Aβ and its value as a biomarker. They found that miR-342-5p decreases the expression of beta-site APP cleaving enzyme 1 (BACE1), Aβ42, and amyloid precursor protein (APP). Opposite results were found with the incubation of exosomes isolated from the hippocampus of a transgenic mouse model of AD compared to wild-type animals. Accordingly, those parameters were ameliorated when artificially loading exosomes with miR-342-5p mimic. In coherence with decreased miR-342-5p signaling during AD, serum exosomes from AD patients show decreased content of miR-342-5p compared to age-matched controls. This is valuable work; however, some improvements can still be performed.

 Major comments,

  1. The calculated fold change for serum exosomes was not normalized but compared to miR-425-3p levels. This is ok as there is no universal loading control for serum exosomes. However, it would be relevant to show how the data behave by normalizing each experiment with miR-425-3p; this could be included as supplementary material.

Response 1: We really appreciate the reviewer’s comments. Some internal RNAs like miR-191-5p, miR-16 and U6 snRNA were widely used as universal loading control for exosomal miRNA analysis (PMC6419325), and miR-425-3p was suggested for the plasma exosomal miRNA profiling (PMC7719248). We then analyzed the miR-342-5p variation trend by the ratio of miR-342-5p to miR-191-5p and miR-342-5p to miR-425-3p. The data implied that miR-342-5p/miR-191-3p was significant downregulated in AD patients compared with that in healthy controls, consistent with the data analyzed with the U6 snRNA as inner control. However, the ratio of miR-342-5p/miR-425-3p showed no significant difference between the two groups, although the value seemed to be lower in AD group than in control (Figure 1).

  1. The table indicating the predicted targets of the miRNA must be somehow categorized. For instance, algorithms for prediction normally give a score for each gene. In addition, it is important to indicate whether the information is validated or not in the literature. Some other works may have validated the downregulation of BACE by miR-425-3p before, if this is the case, it must be indicated.

Response 2: Thank you for the valuable comments. The predicted target genes of miR-342-5p were categorized and listed in Table 3 for the neural function. In fact, we employed TargetScan7.1 bioinformatics tool which retrieved 4127 genes. However, the listed genes in table3 were not validated EXCEPT Bace1 in our work. Furthermore, we performed the TargetScan to predict the targets of miR-425-3p, and among the retrieved data, miR-425-3p may only target Smpd3 (sphingomyelin phosphodiesterase 3, neutral). After searching the PubMed, most of the retrieved papers focused on the roles of miR-425-3p for tumor and therapy. A recent work (PMC9599289) revealed that miR425-3p may be an exosomal biomarker for detection of pancreatic ductal adenocarcinoma (PDAC). Meanwhile, it was demonstrated miR-425-3p was marker for antidepressant response (PMC5477510).

  1. The number of replicates for the western blots in figure 2 must be In addition, the authors should clarify why the relative levels of protein are far from value 1 in the quantifications of control conditions (inhibitor NC and mimic NC). The same applies to figures 3O and 4d

Response 3: We really appreciate the reviewer’s valuable comments. The Western blot experiments were performed in triplicate. We have presented the statement in the manuscript in figure legends of Figure 2, 3 and 4. Although we the authors performed the Western blot assay with equal amount of protein, the cell culture status, protein purification efficiency, SDS-PAGE, and antibody activity would certainly influence the protein detection results in WB data. We performed the WB experiments in triplicate, and analyzed the bands by Image J software. The results revealed that miR-342-5p inhibitor and mimics modulated the protein levels of APP, BACE1 and amyloid beta (showed significant difference in the protein levels according to statistical analysis).

  1. To better show internalization, I would rather display the complete z stack of the confocal pictures. At least, I would show the upper, middle, and lower representative images.

Response 4: We really appreciate the reviewer’s valuable comments. We are sorry for the clerical error when describing the fluorescent experiment. In fact, the fluorescent images were taken by a normal fluorescent microscope, not a confocal fluorescent microscope. We have corrected the sentences in section 2.8, 3.3 and 3.4, and the corresponding figure legends.  

  1. The discussion has a lot of pitfalls. For instance, the statement “These data imply that APP processing and Aβ formation occur in early endosome depending on endocytosis of circulating exosomes” is not well-supported. Similarly, Figure 5, showing the Schematic figure depicting the possible mechanisms underlying the protective effect of exosomal miR-342-5p in Aβ pathology, has an erroneous interpretation of the cellular mechanisms of exosome endocytosis. I strongly suggest correcting the discussion and removing the schematic figure.

 Response 5: We really appreciate the reviewer’s suggestions. We deleted the schematic figure 5 and the corresponding explanations according to the suggestion.

  1. In addition, the discussion does not make completely clear what would be the translational impact of these results when considering the literature showing that decreasing BACE1 does not ameliorate the severity of the disease.

Response 6: We really appreciate the reviewer’s suggestions.

Exosomes are generally engulfed into recipient cells through three different ways: endocytosis, direct membrane fusion, or receptor-ligand interaction. After endocytosis, exosomes subsequently merge with endosomes to release their cargo for signal modulation or be transferred to lysosomes for degradation. Exosomal mRNAs were translated into functional proteins in the recipient cells, suggesting that exosomal mRNAs retain their function in recipient cells. Meanwhile, the exosome miRNAs that target specific genes (in mRNAs status) modulate their expression.

Aβ is derived from amyloid precursor protein (APP) which enriched in neurons. As demonstrated previously (PMID16837572), β-cleavage of APP occurs in early endosomes and Aβ is released in association with exosomes. Noteworthy, as stated in our discussion section, BACE1 location in the early endosomes is critical for the APP processing. Endosomally targeted BACE1 inhibitor specifically blocked the APP cleavage in many cell systems, but not non-amyloid substrates (PMID26923602, reference 45). Also, faster trafficking of BACE1 from early endosome to recycling endosome reduced APP processing and Aβ production (PMID29142073, ref 47). Then we speculated that bioactive molecules in exosomes that modulate BACE1 function in the recipient cellular endosomes might be a promising strategy for reducing Aβ production, possibly without the adverse effect on normal physiological roles of BACE1 and Aβ in other cellular compartments.

Round 2

Reviewer 1 Report

Can be accepted in present form.

Reviewer 2 Report

Although the Authors did not satisfy my request which, in my opinion, would have enriched the paper with a few experiments, the manuscript can be published in the current format.

Reviewer 3 Report

Most of the comments have been answered to an acceptable extent.

Congratulations to the authors.